# Functions of Astrocytes under Normal Conditions and after a Brain Disease

**DOI:** 10.3390/ijms24098434

**Published:** 2023-05-08

**Authors:** Soraya L. Valles, Sandeep Kumar Singh, Juan Campos-Campos, Carlos Colmena, Ignacio Campo-Palacio, Kenia Alvarez-Gamez, Oscar Caballero, Adrian Jorda

**Affiliations:** 1Department of Physiology, School of Medicine, University of Valencia, Blasco Ibañez 15, 46010 Valencia, Spain; 2Indian Scientific Education and Technology Foundation, Lucknow 226002, India; 3Faculty of Nursing and Podiatry, University of Valencia, 46010 Valencia, Spain

**Keywords:** astrocytes, inflammation, learning, memory, neurons, neurodegeneration, neurodevelopment diseases, oxidative stress, sleep disorders

## Abstract

In the central nervous system (CNS) there are a greater number of glial cells than neurons (between five and ten times more). Furthermore, they have a greater number of functions (more than eight functions). Glia comprises different types of cells, those of neural origin (astrocytes, radial glia, and oligodendroglia) and differentiated blood monocytes (microglia). During ontogeny, neurons develop earlier (at fetal day 15 in the rat) and astrocytes develop later (at fetal day 21 in the rat), which could indicate their important and crucial role in the CNS. Analysis of the phylogeny reveals that reptiles have a lower number of astrocytes compared to neurons and in humans this is reversed, as there have a greater number of astrocytes compared to neurons. These data perhaps imply that astrocytes are important and special cells, involved in many vital functions, including memory, and learning processes. In addition, astrocytes are involved in different mechanisms that protect the CNS through the production of antioxidant and anti-inflammatory proteins and they clean the extracellular environment and help neurons to communicate correctly with each other. The production of inflammatory mediators is important to prevent changes in brain homeostasis. On the contrary, excessive, or continued production appears as a characteristic element in many diseases, such as Alzheimer’s disease (AD), amyotrophic lateral sclerosis (ALS), multiple sclerosis (MS), and in neurodevelopmental diseases, such as bipolar disorder, schizophrenia, and autism. Furthermore, different drugs and techniques have been developed to reverse oxidative stress and/or excess of inflammation that occurs in many CNS diseases, but much remains to be investigated. This review attempts to highlight the functional relevance of astrocytes in normal and neuropathological conditions by showing the molecular and cellular mechanisms of their role in the CNS.

## 1. Introduction

Neurons and glia (astrocytes, radial glia, oligodendroglia, and microglia) are the neural cells of the CNS. Glial cells have different functions; microglia are the resident macrophages in the CNS [1,2,3], oligodendrocytes are responsible for myelin production [4], NG2-positive glia is consistent with an oligodendrocyte progenitor function [5,6], and astrocytes play an important role both in homeostasis and in diseases [7,8,9]. Rudolf Virchow indicated that neuroglia was a connective tissue that embedded all the components of the central nervous system (CNS) [10,11,12]. The morphology and functionality of neuroglial cells were established in the 19th and early 20th century [13,14,15]. In addition, analysis of human evolution during the early 21st century revealed the role of neuroglia in the formation of the human intellect [16,17]. Likewise, the involvement of neuroglia in synaptic regulation and plasticity has been demonstrated, generating transient and even lasting changes in the force that can occur in a synapse [18,19]. At the synapse, astrocytes are involved in the regulation of ionic balance, neurotransmitter clearance and gliotransmitter release and they are involved in the maintenance of the blood–brain barrier (BBB), ion and water homeostasis and immune functions, allowing constant regulation of synapses [8,20,21,22]. Additionally, when functional connections fail, microglia and astrocytes remove dendritic processes and spines. Other glial cells, such as oligodendrocytes, form myelin sheaths in the CNS and regulate the speed of information propagation in the CNS, providing conduction plasticity, speed and efficiency in the exchange of information between neurons in a circuit [23]. In the CNS of Homo sapiens, neuroglia cells are predominant and within them, astrocytes constitute the main cells [24], which maintains the homeostasis of the CNS and failure in their functions causes different pathologies in the CNS (Figure 1).

In our manuscript, we show that inflammation is an important mechanism in Alzheimer’s disease. Though previous authors indicated that inflammation is characteristic of this disease, the number of proteins involved and the strong relationship with neurons have been poorly studied. Many chemokines and cytokines are involved, and the relationship between these and the different cells inside the CNS could explain the development of many neurodegenerative diseases. Furthermore, the different roles that astrocytes play in disease remain a priority for many researchers.

The importance of this work lies in showing the functions that astrocytes fulfill in the nervous system. This manuscript introduces the idea that astrocytes participate in memory in a notable way, even in neurodegenerative diseases, causing an affectation in the amnesic cognitive deterioration of patients. In addition, it is indicated that the mechanisms and understanding of the relationship between all cells of the CNS could help to achieve a therapeutic pathway in neurodegenerative and developmental disorders. Moreover, in this review we will discuss how astrocytes contribute to CNS functions owing to their structural properties and their specific functions. This review is not exhaustive and tries to highlight the current state of the field.

## 2. Astrocyte Properties and Functions

Astrocytes contact different populations around neurons, and this is true for protoplasmic astrocytes, which normally contact neurons in the gray matter, and for fibrous astrocytes that contact neuronal extensions [25]. The functions performed by astrocytes are, among others, to control the sleep process, form the extracellular matrix, serve as a support, build and regulate the blood–brain barrier, maintain the balance of extracellular ions, control the production of neurotransmitters, and modulate the synapse (tripartite synapse hypothesis) [26,27]. Memory consolidation occurs during the sleep period and astrocytes can control this process by communicating with other brain cells. During the sleep period they clean toxins, neurotransmitters and release water, ions, and molecules [28].

We believe that neurons and astrocytes are the essential elements in all regulating the processes related to memory of lived situations or dreams. In the CNS, only neurons can produce electrical impulses and therefore help in memory formation and storage. Just like when we turn off a computer and turn it on again after two years, the information stored in it remains, memory created via the communications between neurons remains. We do not know how long-term memory is produced and how memories are maintained but it is established that neurons need to be active all the time to carry out electrical impulses for that memory to be created and stored. Does this imply that we need to have neurons firing constantly if we want to remember something? Astrocytes modulate Ca^2+^ variations and it is possible that using other pathways and signals they are able to communicate with each other and with neurons over time [29]. In fact, in neurodevelopmental disorders, such as autism [30], different forms of schizophrenia [31], and early onset bipolar disorder [32], the number and morphology of astrocytes changes from one situation to another. In bipolar disorder, patients have an increase in brain functions when they are in the optimistic phase and, when they are in the depressive phase they almost find it impossible to register and develop any work in the brain. The change from an optimistic situation to that of depression is not prompt; it takes days, weeks and/or even months, but it is constant. The functions of the neurons cannot explain the above phenomenon since they are cells that at times are activated or at times not (the cell fires or not), and the modulation of the neuronal action potential is impossible. On the contrary, there is a network between the astrocytes and when necessary, a great communication between them and with the neurons, oligodendroglia and microglia is initiated, which is maintained over time [29]. In the evolution of Homo sapiens, the increase in size and complexity of astrocytes coincided with an increase in intelligence [16]. Furthermore, the difference between humans and rats found is the diameter of the average domain of a protoplasmic astrocyte (2.5 times larger), the volume (bigger in humans) and the arborization. Moreover, fibrous astrocytes are 2.3 times larger in humans than in rats [16,33,34].

Currently, a new field has been opened for the study of the role of astrocytes in the nervous system and their possible role in memory. This could provide a new direction for future interventions in CNS diseases.

## 3. Astrocytes Functions and Type of Processes

Astrocytes respond to changes in their microenvironment because they have two types of processes: on one hand, perisynaptic processes, with fine extensions to the interneuronal synapse and, on the other hand, vascular processes with final lengthenings (endfeet) communicating with blood vessels. Perisynaptic processes express receptors for neurotransmitters (reuptake glutamate, GABA and glycine), cytokines, chemokines, growth factors, and ion channels. In addition, they possess the glutamate transporters and receptors that control neuronal glutamatergic neurotransmission [35,36]. The endfeet in astrocytes expresses glucose transporters and aquaporins 4 and covers most of the blood vessels [36]. Moreover, astrocytes are territorial cells with processes controlling only a few overlaps between neighboring astrocytes [37], which are interconnected into functional networks [16,35]. It has been demonstrated that astrocyte cells maintain contact with up to 2,000,000 synapses [17,29] and this interaction depends on changes in neuronal activity. Furthermore, these astrocytes offer energy to neurons via lactate shuttle [38,39] and modulate Ca^2+^ variations [40] (Figure 2). Moreover, they protect neurons from excitotoxicity and help the neuronal differentiation. Astrocytes regulate cellular function controlling extracellular matrix (ECM), cell migration, cell adhesion and cell growth. Furthermore, they control vascular development by producing vascular endothelial growth factor (VEGF) and modulating production of arachidonic acid (AA), nitric oxide (NO) and prostaglandin (PG).

## 4. Toxic Clean Process by Astrocytes

Astrocytes change after inflammation or injury, thus becoming reactive. In this case, there are two types of astrocytes, hypertrophic astrocytes and those that form scars [35,41]. The changes affect different pathologies, such as Alzheimer’s disease (AD), Huntington’s disease, ischemic stroke, and epilepsy. In AD and multiple sclerosis (MS), a reduction in the brain’s weight is detected, with an increase in the grooves and loss of water in general. Astrocytic cells are known to swell with fluid during the day, which it sheds at night, cleaning the brain of toxins, proteins, and unwanted molecules. When the astrocytes are damaged, it is impossible to return water into the astrocytes, thus losing the ability to clean toxins during the wake period and increasing the presence of toxins in the following night [28].

## 5. Glutamine, Glutathione and Astrocytes 

Another function of astrocytes is to supply glutamine to neurons and remove glutathione from the synaptic bouton. In AD, neurons die because of the hyperphosphorylation of TAU (protein π) (pTAU) protein. Astrocytes are probably involved in this process because glutamine reduction and/or glutathione removal may be affected in these patients. In addition, a reduction in ATP produced by damaged mitochondria is detected [28].To function adequately, our brain requires oxygen, which flows through the circulatory system to the extracellular space between brain cells. If the heart function is correct, enough oxygen goes into the brain crossing the blood–brain barrier. On the contrary, if enough oxygen does not go into the brain, problems in brain cell functionality will occur with decrement in the production of adenosine triphosphate (ATP). So, cardiology specialists, neurologists, psychologists, and medical specialists must remain vigilant from now on and in the future while diagnosing AD [42]. Moreover, since astrocytes are involved in non-physiological pain [43], discerning the communication between the cells of the sensory ganglia could be important in the treatment of chronic pain [44,45] (Figure 3).

## 6. Gender Differences and Neuroglia

Regarding gender differences, data analyzed in rodents have shown significant differences in the glia between males and females [46,47]. In the male preoptic area, astrocytes are more ramified, with higher dendritic spine density. They also have a higher density of microglia, which has a reduced branching pattern. As compared to females, males are at a higher risk of developing neurodevelopment disorders, such as autism, different forms of schizophrenia and early onset bipolar disorder, and the role of glia in all of these disorders have already been established. Autism is four to eight times more common in males [48,49] with hyper-masculinized phenotypes [50] (Figure 4).

## 7. Astrocytes and Inflammation

Analogous to microglia, the role of astrocytes in inflammation has been studied. While these cells protect against cerebral ischemia, they appear to be ineffective against lipopolysaccharide-mediated inflammation (LPS). However, in retina cells, astrocytes have been demonstrated to induce an anti-inflammatory and neuroprotective effect through the production of lipoxins against acute and chronic lesions [51]. Furthermore, the cytokine IL-33 produced by astrocytes has an essential role in the development of neuronal circuits [52]. Other studies showed that activation of certain transcription factors is involved in producing protective (STAT3) [53] or injurious effects (NF-ᴋB) [54]. Moreover, there is a correlation between IL-1α and the greater number of GFAP-immunoreactive astrocytes [55]. On the other hand, in multiple sclerosis, TNF-α alters synaptic transmission, affecting the cognitive level [56].

Inflammatory signals are present in patients with mild cognition impairment (MCI) before they develop AD [57]. Inflammation is a crucial factor in AD progression as it is seen in activating microglia and increasing reactive astrocytes in these patients. Astrocytes can change their shape during hypertrophy and increase their ramifications, moving to the injury site [58]. Patients with AD present reactive astrocytes, as detected with PET (Positron Emission Tomography) imaging [59,60] and before the formation of plaques in APP transgenic mice [61].

In reactive astrocytes, the level of gliotransmitters (including glutamate, ATP, d-serine, and GABA) can inhibit neuronal activity [62]. In amyloid plaques, an increase in GABA protein has been detected in reactive astrocytes which surround the plaques and that triggers more release into the extracellular space [62,63]. There is a consensus that the role of GABA is protecting neural cells in the brain [63]. Furthermore, Delekate’s group showed that astrocytes in APP/PS1 mice (Mice carrying the human Swedish amyloid precursor protein and the Δe9 presenilin 1 mutation) increase the release of ATP surrounding the plaques. This happens because the Ca^2+^ concentration rises inside the cell [64]. The latter gives us the idea that an increase in ATP in astrocytes and neurons could help to reduce neuronal death that occurs in AD. The increase in the production of ATP by the mitochondria could help to recover from AD and decrease the development of the illness. The use of a cold laser could elevate the ATP produced by mitochondria as it can act on cytochrome c oxidase, increasing energy in the cells. This could help patients with mild cognitive impairment before they progress into AD [42]. However, in AD patients, the use of the laser could be unproductive in many cases because the cells are already dead. MS is a CNS (Central Nervous System) disorder characterized to be a chronic inflammatory autoimmune disease. Both environmental and genetic factors [65] are involved, and patients develop three states:(1)Presenting a relapsing–remitting clinical course, characterized by episodes of acute neurological dysfunctions followed by periods of recovery [66] (20–30% of patients).(2)Progression to a chronic secondary clinical stage, that is characterized by worsening and increased disability (50% of patients) [67].(3)In 15% of the patients, MS progresses to the clinical gradual reduction in neurological functions [68].

Animal models of MS disease show that it is an autoimmune inflammatory disorder occurring due to the recruitment of reactive lymphocytes, CD4^+^ T cells (Cluster of Quadruple Differentiation in T cells and the dendritic cells). Focusing on the three states, it seems that in the first state the bipolar relapsing and remitting states look similar to the illness produced by the virus chikungunya (CHIK) [69], or the plasmodium (as malaria) or a bacterial infection, with higher or lower episodes of affectation. The bipolar action will be related to the immune cells and their role in controlling the damage in different diseases.

The gliotransmitter glutamate is released by astrocytes when beta-amyloid is detected in the medium and causes loss of the neural spine and synaptic damage due to activation of NMDA (N-Methyl-D-Aspartate) receptors [70]. In addition, astrocytes release purines that can influence the development of AD or other diseases and activate the production of inflammatory proteins, decreasing anti-inflammatory proteins [71]. In a 7-month-old APP/PS1 mice, an increase in chemokines and their receptors has been detected compared to wild-type mice, demonstrating the role of cytokines and chemokines produced by microglia and astrocytes [71,72]. In addition, Valles’ group detected, for the first time, changes in the expression of CCR5 (Chemokine (C-C motif) Receptor 5) and CCR8 (Chemokine (C-C motif) Receptor 8) in these mice with a high production of β-amyloid [72]. These results demonstrate that changes in inflammatory protein expressions could affect neurodegeneration, brain development, cognition and memory [71,73]. Furthermore, using lipopolysaccharide (LPS) as an inducer, astrocytes increase the expression of many genes in the complement cascade [74]. On the other hand, the upregulation of trophic factors after ischemic damage has been shown to be a protective mechanism. However, the role of astrocytes could be different depending on the age and situation of these cells in our body and there are probably not only two states of astrocytes. Neurotrophic factors (NTF), and their action on neuronal survival have led to the search for new treatments for neurodegenerative diseases. One of them is the mesencephalic astrocyte-derived neurotrophic factor (MANF). This treatment is unique because its amino acid sequence and three-dimensional structure is different from other NTFs. MANF is secreted after stress or cell injury to promote neuronal survival. Its mechanism of action is unknown as well as its receptors. MANF is a neuroprotective protein with great therapeutic potential for PD [75] and AD [76]. PD is characterized by tremors, rigidity and bradykinesia, caused by the death of dopaminergic neurons in the substantia nigra of the brain. MANF protects against cell death through the PI3K/Akt/GSK3β pathway and by upregulating stress genes, such as HSP70 [77].

## 8. Astrocytes and Oxidative Stress

Free radicals are pro-oxidant molecules that contain one or more unpaired electrons. Depending on the molecules, free radicals can come from oxygen, nitrogen, lipids, etc. Furthermore, they can take electrons from other molecules, making them highly reactive [78]. The increase in ROS levels (Reactive Oxygen Species) is related to many neurodegenerative diseases [79,80]. The effects of antioxidants in clinical studies have been very disappointing because the high concentration of antioxidants acts in many cases as a pro-oxidant [81], because oxidative stress occurs relatively early in the course of diseases, and/or the combination of antioxidants falls into a clinical situation [82]. Activated immune cells can produce ROS that contributes to mitochondrial dysfunction and neural cell death by apoptosis (Figure 5). Superoxide dismutase (SOD) converts superoxide free radicals into molecular oxygen and hydrogen peroxide, which are broken down by the enzyme catalase [83,84]. Microglial cells are the primary immune cells in the CNS and the role of oxidative stress has been well studied in this cell type. In contrast, studies on the role of astrocytes in this process is scarce. Astrocytes protect neurons from oxidative stress by producing antioxidant proteins [41].

The toxic peptide beta-amyloid causes the production of hydrogen peroxide by astrocytes [85], as it has been indicated before [80], and these ROS are released in response to beta-amyloid through the pentose phosphate pathway. Drugs that can protect astrocytes and neurons from inflammation and/or oxidative stress damage have been used in AD cells in culture, in animal models and in humans [73,86,87], but none of them have obtained a recovery and/or less development of AD. The results are the same in studies conducted for other diseases.

Due to the high metabolic rate of neurons, these cells produce more free radicals than other cells in the nervous system. Neurons also have a reduced ability to eliminate reactive oxygen species, which makes them highly vulnerable to oxidative stress [88]. In many neurodegenerative disorders, such as PD, AD, ALS, HD, etc., an increase of free radicals that can induce aggregations of proteins has been detected. [89,90,91,92]. In sporadic cases of Alzheimer’s disease, age factors are important and can influence the Aβ-amyloid processing by weakening mitochondrial function and producing excessive ROS [93]. Alternatively, α-synuclein aggregates caused by dopaminergic neurons in Parkinson’s disease disrupt mitochondrial function and then produce oxidative stress [94,95]. Moreover, α-synuclein aggregation can be produced by an increase in oxidative stress [90]. One form of amyotrophic lateral sclerosis (ALS) caused by superoxide dismutase 1 (SOD1) mutation [96], presents the affectation of the antioxidant molecules balance with the increase in oxidative stress [97] due to aggregation of SOD 1 [98]. The diseases indicated above, along with HD, share aggregation processes and oxidative stress that together engender a vicious cycle [99].

## 9. Neurodegeneration Mechanisms

In the neurodegenerative cascade, several basic mechanisms can join in, such as apoptosis, necrosis, autophagy, retrograde neurodegeneration, Wallerian degeneration, demyelination and astrogliopathy [100]. There is evidence of apoptotic mechanisms in animal models of various neurodegenerative diseases, but the evidence in human tissues is limited. The activation of caspase 1, 3, 8 and 9 and the release of cytochrome c observed in models of Huntington’s disease (HD) is demonstrated in human striated brain tissue [101,102]. Similarly, caspases activation and neuronal apoptosis have been detected in ALS [103] and HIV [104]. In necrosis, with non-caspases-dependent death, two major effector proteins act, serine/threonine-protein kinase 1 (RIPK1) and the mixed lineage kinase domain (MLKL). In murine ALS models, release of TNF-α, FasL and TRAIL by astrocytes has been detected that can trigger necrosis through the activation of RIPK1 and MLKL [104]. Furthermore, in humans with ALS, a normal pathology mediated by RIPK1 has been detected [105]. On the other hand, in MS, necrotic mechanisms are also observed in pathological samples [106]. 

Apathy has multifactorial symptoms, such as behavioral, cognitive, and emotional facets including impaired motivation and reduced goal-directed behavior. Apathy belongs to schizophrenia, bipolar disorders, and autism’s negative symptomatology, although the molecular mechanisms are still poorly studied [107,108]. Correlations between apathy with specific brain regions and executive functions have been shown (the anterior cingulate cortex, orbitofrontal cortex, and the ventral and dorsal striatum). It is considered the major neuropsychiatric symptom in both acquired and neurodegenerative disorders, such as strokes [109], AD [110], ELA [111] or Parkinson’s disease [112]. All these disorders have a disturbance in the normal balance of neurotransmitters and are associated with anomalies in specific brain regions and inflammatory pathways leading to glia activation and finally neuronal and neural loss [113]. In MS there is a decomposition of the blood–brain barrier (BBB), death without regeneration of oligodendrocytes, loss of myelin, axonal degeneration and reactive gliosis of astrocytes and activation of microglia [114,115]. In this disease, inflammation plays an important role by increasing in cytokines and chemokines. In the pathophysiology of MS, the BBB is compromised, causing the activation of the microglia and the immune cells of the periphery. The microglia not only produces pro-inflammatory cytokines and chemokines secretion with decreased anti-inflammatory agents but also releases reactive oxygen and glutamate species [116]. Each type of cell of the innate and adaptive immune system can organize the inflammatory response within the CNS and the autoreactive CD4 + T cells make an important contribution in the MS.

## 10. Astrocytes, Sleep Process and Diseases

A century ago, Santiago Ramón y Cajal proposed astrocytes as cells that regulate the process of sleep [28]. He detected several processes in the synapses during sleep and observed retraction of those processes during wakefulness. In 2009, scientists began to take an interest in sleep and the role of astrocytes in it and discovered that the influence of astrocyte on the sleep/wake cycle [117] and then, astrocytes were postulated as modulator members of the homeostasis of the sleep cycle [118]. Astrocytes release adenosine units to its receptor, adenosine A1, occasioning sleep and driving to total sleep [119]. Furthermore, it is known that astrocytes clean the brain during sleep, releasing solutes and water inside the brain, cleaning it through astrocytic aquaporins 4. In addition, in vivo microdialysis studies have shown that β-amyloid (the toxic peptide in Alzheimer’s brain) increases within the interstitial fluid during wakefulness and decreases during the sleep process [119]. Thus, the brain cleaning that occurs during the sleep period is decreased in Alzheimer’s patients compared to control individuals [120,121]. In addition, the changes observed in the amount of toxic peptide decrease during the development of AD [122], so detection in the glymphatic pathway could be affected in neurodegenerative patients and in other diseases, such as bipolar disorders, chronic fatigue syndrome, MS and schizophrenia [123]. Furthermore, the sleep/wake cycle, modulated by astrocytes, is also altered in those diseases [65,124,125]. Many people suffering from these diseases present a REM (Rapid Eye Movement) alteration process during the sleep period. This data could indicate alterations in the sleep period of the active zone of the brain that initiates the off-switch in the brain. In fact, the patients cannot sleep well, and they feel tired during the awake period, with problems in attention, memory, spatial recognition, and so on.

## 11. Therapeutic Effects to Combat Diseases

At present, there are many therapeutic drugs used against diseases of the nervous system, however, it is also important to test the use of new drugs and approaches against those diseases to treat them more effectively (Table 1). Future therapeutics against brain diseases must develop specific drugs against reactive astrocytes and microglia activation. Studies on the mechanisms that eliminate amyloid beta toxic peptide, the decrement of the pTAU inside the neurons, the ATP changes in the brain controlled by astrocytes and the production of metabolites, will be necessary for finding therapeutic targets in AD and other diseases [126].

Crises due to sensory overstimulation in people with autism are involuntary. The body of a person with autism react to sensory stimuli and the person being unable to bear the overstimulation suffers from a nervous breakdown. Some approaches could produce beneficial physiological effects, such as the reduction in deep pressure to achieve a decrement in anxiety in children with autism spectrum disorders (ASD) [127,128]. Furthermore, the use of new drugs, such as monoclonal antibodies designed to elicit an immune response to eliminate senile plaques, which damage communication between brain cells and end up killing the neurons, will be necessary. In chronic pain, drugs controlling the mechanisms of SG cells and the interaction of these cells with the neuronal body will be important to assume the relationship between astrocytes and the other cells in the sensorial ganglia and chronic pain treatment. In the future, for better health, brain changes in sleep-inducing proteins and the sleep/wake cycle could help fighting sleeping disorders and neurodegenerative diseases as well. In addition, the influence of astrocytes in the brain cleaning process will be a therapeutic approach to eliminate the toxic elements detected in many diseases, such as in AD, Parkinson’s, or ALS (Amyotrophic Lateral Sclerosis) and many other neurodegenerative diseases in which toxic proteins are present. Moreover, altered mRNA expression profile in ALS and/or other neurodegenerative diseases has been detected with an increase in inflammation produced by microglia and astrocyte reactivities, which are potential mediators of neurodegenerative processes [97,129].

**Table 1 ijms-24-08434-t001:** Drugs and their effects on glia.

DRUGS	EFFECT ON ASTROCYTES	REFERENCES
IFN-β	Downregulation of cytokine, NO production, and MMP generation	Bhat et al., 2019 [130]
Pioglitazones	Inhibit inflammation reducing glial activation	Dhapola et al., 2021 [131]
Fingolimod	Immune-modulator inhibitor actions on activated microglia	Pitteri et al., 2018 [132]
Minocycline	Immune-modulator inhibitor actions on activated microglia	Wang et al., 2020 [133]
Xaliproden	Production of increased amount of neurotrophic factors	Lacomblez et al., 2004 [134]
Cyclooxygenase (COX) inhibitors	Improved memory and decrease amyloid deposition	Lopez-Ramirez et al., 2021 [135]
Propentofylline	Decrement of neuroinflammation relative to glial cell activation	Sweitzer and De Leo, 2011 [136]
Glatiramer acetate	Anti-inflammatory cytokine and neurotrophic factors increase	Kasindi et al., 2022 [137]
Verapamil + magnesium sulphate	Block overactivation of L-Type calcium channels	Zhang et al., 2019 [138]
Rapamycin	Astrogliosis inhibition	Selvarani et al., 2021 [139]
Fuoxetine	Decrement of antigen-presenting	Barakat et al., 2018 [140]
Teriflunomide	NO synthesis downregulation	Hauser et al., 2020 [141]
Methotrexate	Induction of astrogliosis, injury	Shao et al., 2019 [142]
Tacrolimus	Inhibition of pro-inflammatory cytokines	La Maestra et al., 2018 [143]
Mycophenolate mofetil	Downregulation of NO synthesis	Ebrahimi et al., 2012 [144]
Glutocorticoids	Dwonregulation of pro-inflammatory cytokines, and astrogliosis	Nichols et al., 2001 [145]

It is necessary to promote the clearance of toxic proteins by astrocytes (such as amyloid beta) via different mechanisms, such as autophagy or ubiquitin systems. In addition, in reactive astrocytes, an increase in antioxidant proteins, such as Nrf2 (Nuclear Factor Erythroid 2-related factor 2), could benefit our brain. On the other hand, astrocytes can clear toxic peptides from the brain leading to reactive astrocytes and increased inflammation, toxic proteins, and oxidative stress molecules. Regulation of the oxidative stress state and inflammation could help neurons located near astrocytes to survive. Additionally, astrocytes can increase GABA levels after damage, so controlling MAO-B (monoamine oxidase-B) activity could help rescue the brain from memory problems, such as those found in the AD. Cerebral blood flow (CBF) decreases with age. Between the ages of 20 and 60, the CBF falls by 16% and continues to fall by 0.4% each year. A reduction in oxygen and glucose supply to the brain occurs, and this drop in CBF reduces ATP energy production. Mitochondrial loss or damage with reduced ATP worsens when vascular risk factors (VRFs) develop during Alzheimer’s disease and may accelerate CBF declination and mitochondrial deficiency where mild cognitive impairment (MCI) develops [42]. One form of photobiomodulation (PBM), transcranial infrared brain stimulation (TIBS), is planned in a randomized, placebo-controlled study of MCI patients which is to be conducted at our university. Photobiomodulation has been used in Parkinson’s disease, depression, traumatic brain injury and stroke with reported benefits. Medical interventions, pharmacological approach, etc. have been used in AD, but TIBS will be a better technique for the future. The study of the effects of photobiomodulation on the brain during aging has been studied and reviewed by many authors [146,147].

The heterogeneity of astrocytes is poorly understood and has not been sufficiently studied. The study of the different forms that astrocytes acquire to activate and their actions during the processes in different neurological diseases must be deepened and has not been sufficiently explained in this manuscript. Nor has it been addressed whether it is possible to act therapeutically on the different types of astrocytes in neurological diseases.

## 12. Conclusions

Controlling mechanisms and understanding the relationship between astrocytes, neurons, oligodendroglia, and microglia could become the therapeutic track to some neurodegenerative disorders and diseases, such as bipolar disorder, schizophrenia, and autistic spectrum. Thus, different strategies can be considered to bridge the gap between human disorders and astrocytes and glia intervention. Furthermore, the presence of different types of astrocytes and increases or decreases in them depending on the disease and age should be a principal focus of studies in the future (Figure 6). Future investigations must be carried out to understand the general mechanisms that produce morphological deficits of astrocytes in neurodegeneration. It is imperative to know if these deficits produce functional changes in astrocytes and if neurodegeneration can be slowed down. Hypothesis looking for the role of astrocytes in mammalian functions will be necessary as a new field in the astrocytes’ job in the nervous system has been opened now. This will provide a new direction for future interventions in CNS diseases.

## Figures and Tables

**Figure 1 ijms-24-08434-f001:**
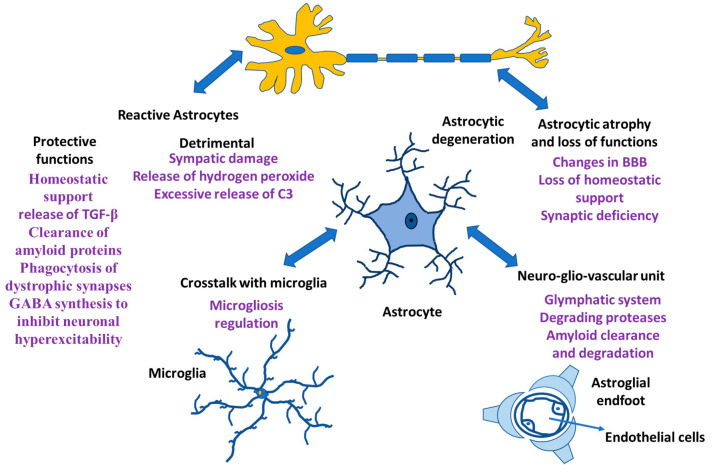
Astrocyte functions. Astrocytes and their relationship with other cells both in physiological situation and in neurological damage.

**Figure 2 ijms-24-08434-f002:**
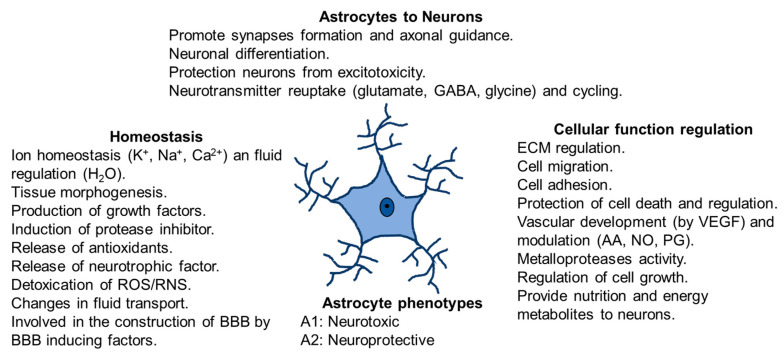
Astrocytes functions in physiological conditions. Homeostasis, communication astrocytes-neurons, cellular function regulation and astrocytes phenotypes.

**Figure 3 ijms-24-08434-f003:**
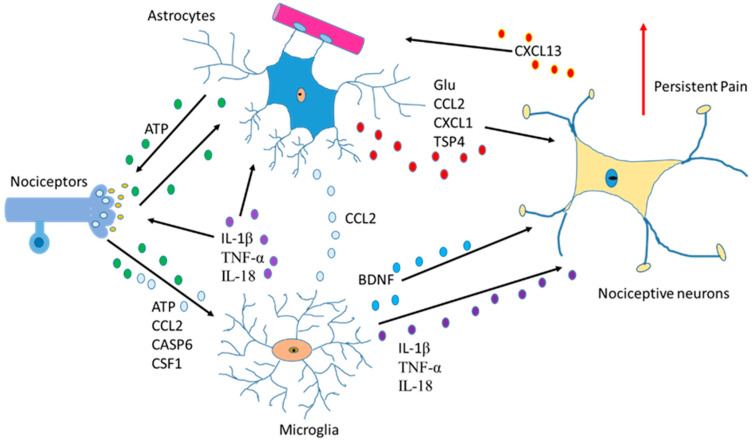
Communication between astrocytes, neurons, and microglia in pain situations. Activation of nociceptors causes the induction of cytokines, chemokines, BDNF and neurotrophic factors producing changes an increase in persistent pain.

**Figure 4 ijms-24-08434-f004:**
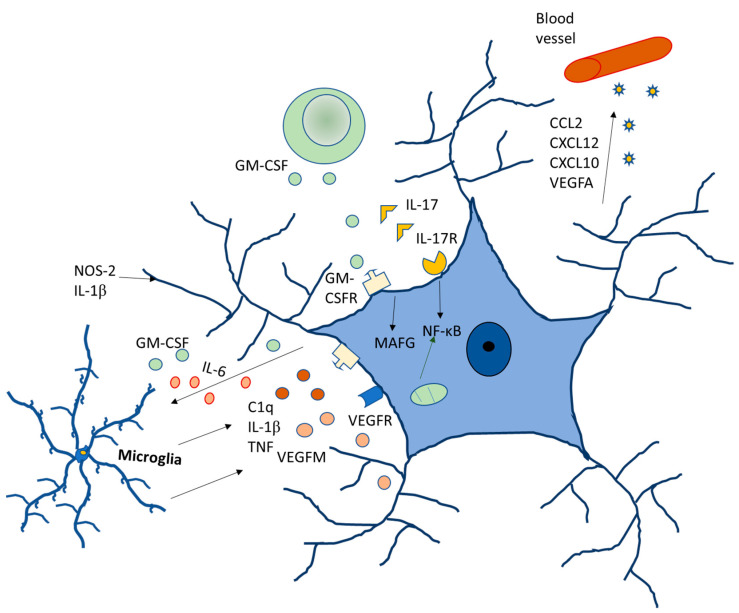
Molecular signaling of glia modulating neurodegenerative diseases.

**Figure 5 ijms-24-08434-f005:**
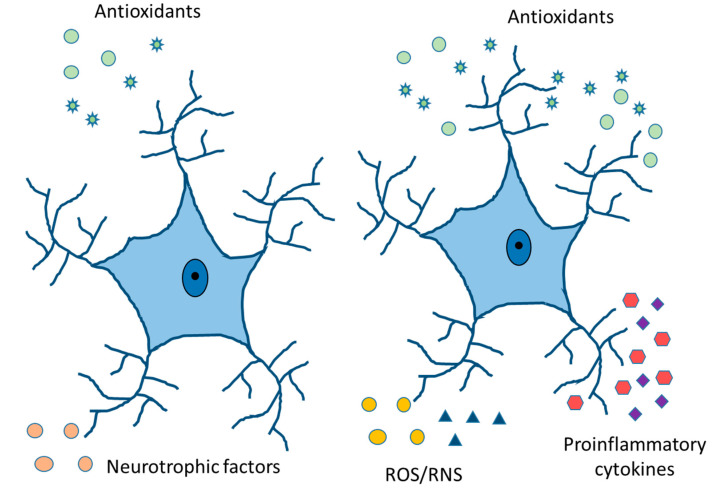
Physiologically healthy and reactive astrocytes. Release of neurotrophic and antioxidant factors, ion homeostasis, ROS/RNS detoxification, fluid transport, vasodilation, and neurotransmitter reuptake in healthy astrocytes. Cytokine and chemokine release, ROS/RNS production, increased expression of GFAP, vimentin, and Glu, and compensatory release of antioxidants in reactive astrocytes.

**Figure 6 ijms-24-08434-f006:**
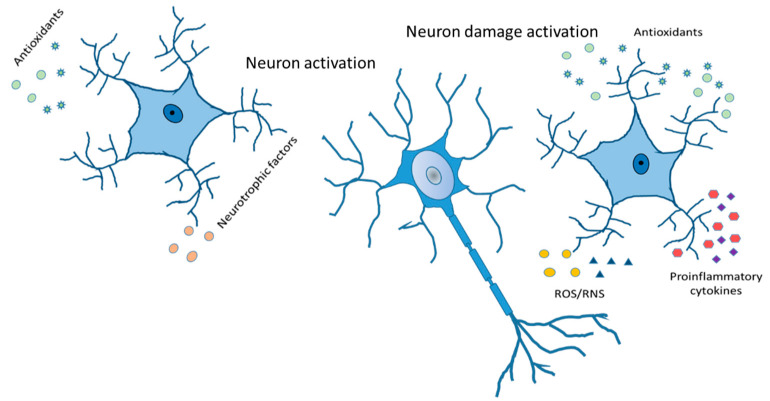
Reactive astrocytes. Release of antioxidants and neurotrophic factor activate neurons in a good way. Cytokines and chemokines, ROS/RNS production, increased expression of GFAP, vimentin and Glu, and compensatory release of antioxidants leading to neuron damage activation.

## Data Availability

No new data was created.

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
