# Peer review of "Functions of Astrocytes under Normal Conditions and after a Brain Disease"

_ijms, 2023, doi:10.3390/ijms24098434_

Round 1

Reviewer 1 Report

The authors reviews important roles of glial and astocytic cells in different brain functions.

I noticed that: 

1- The manuscript on a formate written as: Int. J. Mol. Sci. 2021, 22, x. https://doi.org/10.3390/xxxxx

2- The closing statement of the abstract should be imporved and not as such vague meaning "This review is intended to serve as an approach to the field:.

3- Many references are obsolete and should be updated "1985, 1996, etc"

4- All genes should be written in italic characters "A basic rule of writing genetic names".

5- Some proteins are active in a phosphorelated status. The authors have not differenciate between proteins in a Protein form and those in a phosphorelated active form. This needs to be revised. (Such as phosphorelated Tau should be written pTau)

Author Response

REFEERE 1

1- The manuscript on a formate written as: Int. J. Mol. Sci. 2021, 22, x. https://doi.org/10.3390/xxxxx

We thank the reviewer for his suggestions to improve the manuscript.

2- The closing statement of the abstract should be imporved and not as such vague meaning "This review is intended to serve as an approach to the field:

The paragraph “This review is intended to serve as an approach to the field.” Has been eliminated and substituted by “This review attempts to show the functional relevance of astrocytes in normal and neuropathological conditions by showing the molecular and cellular mechanisms of the role of astrocytes in the CNS”. We hope the reviewer finds this change appropriate.

3- Many references are obsolete and should be updated "1985, 1996, etc"

We changed the obsolete articles by others the more actuality.

4- All genes should be written in italic characters "A basic rule of writing genetic names".

We changed the genes with italic characters, as reviewer suggest.

5- Some proteins are active in a phosphorelated status. The authors have not differenciate between proteins in a Protein form and those in a phosphorelated active form. This needs to be revised. (Such as phosphorylated Tau should be written pTau)

We add the active phosphorylated form of the protein.

We hope that we have responded adequately to all the indications that the referee gave us.

Reviewer 2 Report

The review manuscript “FUNCTIONS OF ASTROCYTES UNDER NORMAL CONDITIONS AND AFTER A BRAIN DISEASE” by Valles et al., addresses the key function of astrocytes in central nervous system under normal verses pathological conditions. Although the authors have drafted the manuscript well, but it still requires extensive English editing and modifications as suggested below to further improve the manuscript quality for consideration for publication.

1.      The title of the manuscript may be reframed as “Functional relevance of Astrocytes under normal and neuropathological conditions”. Also, explain the rationale and importance of the present review compared to the similar reviews already published.

2.      The source for the figures 1, 3 and 5 need to be mentioned in the legends and same need to be incorporated in the text. Also, it is necessary to include it in the current review manuscript?

3.      Figure 6 and 7 may be merged as one figure.

4.      What is the rationale for including the figure 2 ? Instead, authors should include a figure depicting at molecular signaling level that how astrocytes modulate neurodegenerative diseases.

5.      Table depicting current drugs in market and trail targeting the astrocytes to ameliorate neuroinflammation should be included in the manuscript.

6.      The generalized figure showing the astrocyte function under normal physiological condition should also need to be incorporated to attract readers for the papers.

7.      Authors needs to more elaborate with current references on molecular interaction between the astrocytes, microglia, and neurons in relevance to neuroinflammation and neurodegenerative outcomes. 

Author Response

We thank the reviewer for his suggestions to improve the manuscript.

the review manuscript “FUNCTIONS OF ASTROCYTES UNDER NORMAL CONDITIONS AND AFTER A BRAIN DISEASE” by Valles et al., addresses the key function of astrocytes in central nervous system under normal verses pathological conditions. Although the authors have drafted the manuscript well, but it still requires extensive English editing and modifications as suggested below to further improve the manuscript quality for consideration for publication.

  1. The title of the manuscript may be reframed as “Functional relevance of Astrocytes under normal and neuropathological conditions”. Also, explain the rationale and importance of the present review compared to the similar reviews already published.

As the referee suggested, we reframed the title of the manuscript as “Functional relevance of Astrocytes under normal and neuropathological conditions”. 

The importance of this work lies in showing the functions that astrocytes play in the nervous system. To introduce the idea that astrocytes participate in memory in a notable way, even in neurodegenerative diseases, causing an affectation in the amnesic cognitive deterioration of the patients. In addition, it is indicated that the mechanisms and the understanding of the relationship between ALL THE CELLS of the CNS could help to achieve a therapeutic track in neurodegenerative and developmental disorders.

  1. The source for the figures 1, 3 and 5 need to be mentioned in the legends and same need to be incorporated in the text. Also, it is necessary to include it in the current review manuscript?

All the figures have been renamed since they do not follow a chronological order and new figures have been added to the text as reviewer suggested. Some of them have been eliminated as suggested the editor (because they are original figures from the Dr. Valles laboratory)

As referee suggested Figure 6 and 7 older has been merged as one figure (Figure 5). We add in the legend: Physiologically healthy and reactive astrocytes. Release of neurotrophic and antioxidant factors, ion homeostasis, ROS/RNS detoxification, fluid transport, vasodilation, and neurotransmitter reuptake in healthy astrocytes. Cytokine and chemokine release, ROS/RNS production, increased expression of GFAP, vimentin, and Glu, and compensatory release of antioxidants in reactive astrocytes.

  1. What is the rationale for including the figure 2 ? Instead, authors should include a figure depicting at molecular signaling level that how astrocytes modulate neurodegenerative diseases.

We have added a new Figure as reviewer suggested.

  1. Table depicting current drugs in market and trail targeting the astrocytes to ameliorate neuroinflammation should be included in the manuscript.

A table was added as reviewer suggested.

  1. The generalized figure showing the astrocyte function under normal physiological condition should also need to be incorporated to attract readers for the papers.

We add a figure with the astrocyte functions in normal physiological conditions.

  1. Authors needs to more elaborate with current references on molecular interaction between the astrocytes, microglia, and neurons in relevance to neuroinflammation and neurodegenerative outcomes.

A new paragraph and table have been added in THERAPEUTIC EFFECTS TO COMBAT DISEASES.

At present, the therapeutic drugs against diseases of the nervous system have been many, but the use of new drugs and approaches to the problem are needed and must be carried out (Table 1).

Furthermore, new figures with molecular interaction between astrocytes, microglia and neurons have been added.

Reviewer 3 Report

1.      Please use a proposer template.

2.      The abstract section should be enhanced to include quantitative data.

3.      At the end of your abstract, please provide a "take-home" message.

4.      Put the keywords in a new order based on alphabetical order.

5.      What is the current article novel? It has been extensively discussed in the past. Nothing truly novel in its current state. The absence of anything original makes the current study seem like a replication or a modified study. The introduction section should contain specifics about the writers' uniqueness. It is a significant reason to reject this study.

6.      In the relation to autism spectrum disorder, please giving additional relevant reverence to support the explanation as follows: Physiological Effect of Deep Pressure in Reducing Anxiety of Children with ASD during Traveling: A Public Transportation Setting. Bioengineering 2022, 9, 157. https://doi.org/10.3390/bioengineering9040157

7.      Previous literature related needs to explain in the introduction section consisting of their work, their novelty, and their limitations to show the gaps that intend to be filled in the present work.

8.      In the last paragraph of the introduction, please explain the objective of the present article.

9.      Recommended to add an additional figure in the introduction section to improve the presentation of the present article.

10.   In order to improve the reader's understanding of the materials and methods section simpler, the authors could provide figures that clarify the workflow of the current study rather than only the predominant text as it currently appears.

11.   What is the baseline of patient selection? Is there any protocol, standard, or basis that has been followed? It is unclear since the patient is very heterogeneous with a small number. The resonance involved impacts the present result making this study flaws. One major reason for rejecting this paper.

12.   Other information about the tool, such as the manufacturer, country, and specifications, should be provided.

13.   Important information that must be mentioned in the publication relates to the error and tolerance of the experimental equipment utilized in this investigation. As a result of the disparate findings in subsequent research by other researchers, it would be a useful discussion.

14.   Results comparison with similar previous studies needs to give.

15.   Overall, the discussion in the present article is extremely poor. The Authors must extend their discussion and make a comprehensive explanation.

Author Response

We thank the reviewer for his suggestions to improve the manuscript.

  1. The abstract section should be enhanced to include quantitative data.

The quantitative data have been added to the abstract section.

  1. At the end of your abstract, please provide a "take-home" message.

At the end of the abstract, we have added.

“This review attempts to show the functional relevance of astrocytes in normal and neuropathological conditions by showing the molecular and cellular mechanisms of the role of astrocytes in the CNS.”

  1. Put the keywords in a new order based on alphabetical order.

The keywords as been changes to alphabetical order.

  1. What is the current article novel? It has been extensively discussed in the past. Nothing truly novel in its current state. The absence of anything original makes the current study seem like a replication or a modified study. The introduction section should contain specifics about the writers' uniqueness. It is a significant reason to reject this study.

The importance of this work lies in showing the functions that astrocytes play in the nervous system. To introduce the idea that astrocytes participate in memory in a notable way, even in neurodegenerative diseases, causing an affectation in the amnesic cognitive deterioration of the patients. In addition, it is indicated that the mechanisms and the understanding of the relationship between ALL THE CELLS of the CNS could help to achieve a therapeutic track in neurodegenerative and developmental disorders.

We add a paragraph in the introduction as reviewer suggest.

  1. In the relation to autism spectrum disorder, please giving additional relevant reverence to support the explanation as follows: Physiological Effect of Deep Pressure in Reducing Anxiety of Children with ASD during Traveling: A Public Transportation Setting. Bioengineering 2022, 9, 157. https://doi.org/10.3390/bioengineering9040157

In the review has been add a new paragraph. “Other approaches could produce beneficial physiological effects such as the reduction of deep pressure to achieve a decrease in anxiety in children with autism spectrum disorders (ASD)”.

A new reference has been added too.

  1. Previous literature related needs to explain in the introduction section consisting of their work, their novelty, and their limitations to show the gaps that intend to be filled in the present work.

We increase introduction as reviewer suggest.

  1. In the last paragraph of the introduction, please explain the objective of the present article.

We did it as reviewer suggest.

  1. Recommended to add an additional figure in the introduction section to improve the presentation of the present article.

We add a new figure in introduction to improve the present manuscript.

  1. In order to improve the reader's understanding of the materials and methods section simpler, the authors could provide figures that clarify the workflow of the current study rather than only the predominant text as it currently appears.

We add a table and new figures to improve the manuscript and more paragraphs  

  1. What is the baseline of patient selection? Is there any protocol, standard, or basis that has been followed? It is unclear since the patient is very heterogeneous with a small number. The resonance involved impacts the present result making this study flaws. One major reason for rejecting this paper.

Patients have been selected at an age between 70 to 80 years. They are patients with mild cognition impairment (MCI) of the amnestic type who have not yet developed Alzheimer's disease. The use of a form of photobiomodulation, transcranial infrared brain stimulation (TIBS), was performed in a randomized, placebo-controlled study of MCI patients, using a standard protocol already published in the laboratory of Dr. Francisco Gonzalez-Lima. This author and our group have already demonstrated that the machine is safe for the patients.

  1. Other information about the tool, such as the manufacturer, country, and specifications, should be provided.

The machine was developed in the laboratory of Dr. Francisco González-Lima and transferred to my laboratory to carry out the pilot study. This study has already been completed and will be published soon.

  1. Important information that must be mentioned in the publication relates to the error and tolerance of the experimental equipment utilized in this investigation. As a result of the disparate findings in subsequent research by other researchers, it would be a useful discussion.

We add this sentence at the end of the manuscript. “The study of the main brain effects of photobiomodulation during aging have been study and reviewer for many authors (127,128)” Furthermore, we have added these two papers.

  1. Results comparison with similar previous studies needs to give.We added new refferences. -Cardoso, F. D. S.; Gonzalez-Lima, F., & Gomes da Silva, S. Photobiomodulation for the aging brain. Ageing research reviews 2021, 70, 101415. -Heiskanen, V., & Hamblin, M. R. Photobiomodulation: lasers vs. light emitting diodes? Photochemical & photobiological sciences: Official journal of the European Photochemistry Association and the European Society for Photobiology 2018, 17, 1003–1017. 
  2. Overall, the discussion in the present article is extremely poor. The Authors must extend their discussion and make a comprehensive explanation.

We add extend the manuscript as reviewer suggest.

Round 2

Reviewer 2 Report

The manuscript is significantly improved after incorporating the reviewers  suggestions. Authors may also include an additional column in Table citing the references for the drugs mentioned in the Table. 

Author Response

The manuscript is significantly improved after incorporating the reviewer’s suggestions. Authors may also include an additional column in Table citing the references for the drugs mentioned in the Table.

As reviewer suggested, a new Table with references has been added.

Reviewer 3 Report

1.      What is the current work's limitation? Please place it before entering the conclusion section.

2.      Express the conclusion in the form of a paragraph rather than in the current form, that is point by point.

3.      Further research needs to be explained in the conclusion section.

4.      Explanation related anxiety in autism spectrum disorder needs to included. Please provide it along with relevant reference as follows: Effect of Short-Term Deep-Pressure Portable Seat on Behavioral and Biological Stress in Children with Autism Spectrum Disorders: A Pilot Study. Bioengineering 2022, 9, 48. https://doi.org/10.3390/bioengineering9020048

5.      Across the article, the author conducted paragraphs that were only one or two phrases long, making the explanation unclear. The authors should elaborate on their explanation to create a more thorough paragraph. It is advised that each paragraph have at least three sentences, with one sentence operating as the primary sentence and the other sentences operating as supporting sentences.

6.      The reference needs to be enriched from the literature published five years back. MDPI reference is strongly recommended.

7.      Please reduce the literature used as a reference that is authored by the present author in order to reduce the number of self-citation.

8.      The authors need to proofread the manuscript due to grammatical errors and language style.

9.      After peer review, it is encouraged that a graphical abstract be included in the submission.

Author Response

  1. What is the current work's limitation? Please place it before entering the conclusion section.

A new paragraph has been inserted before the conclusion section as indicated by the reviewer.

The heterogeneity of astrocytes is poorly understood and has not been sufficiently studied. The study of the different forms that astrocytes acquire to activate and their actions during the processes in different neurological diseases must be deepened and has not been sufficiently explained in this manuscript. Nor has it been addressed whether it is possible to act therapeutically on the different types of astrocytes in neurological diseases.

  1. Express the conclusion in the form of a paragraph rather than in the current form, that is point by point.

Changes have been made as indicated by the author.

  1. Further research needs to be explained in the conclusion section.

A new paragraph has been inserted in the conclusion section as indicated by the reviewer.

Future investigations must be carried out to understand the general mechanisms that produce morphological deficits of astrocytes in neurodegeneration. Furthermore, to know if these deficits produce functional changes in astrocytes and if neurodegeneration can be slowed down.

  1. Explanation related anxiety in autism spectrum disorder needs to included. Please provide it along with relevant reference as follows: Effect of Short-Term Deep-Pressure Portable Seat on Behavioral and Biological Stress in Children with Autism Spectrum Disorders: A Pilot Study. Bioengineering 2022, 9, 48. https://doi.org/10.3390/bioengineering9020048

We include the explanation related anxiety in autism spectrum disorder and the reference indicated by the reviewer.

Crises due to sensory overstimulation in people with autism are involuntary. The body reacts to sensory stimuli and the person with this disease cannot bear the overstimulation and breaks down in a nervous breakdown.

  1. Across the article, the author conducted paragraphs that were only one or two phrases long, making the explanation unclear. The authors should elaborate on their explanation to create a more thorough paragraph. It is advised that each paragraph have at least three sentences, with one sentence operating as the primary sentence and the other sentences operating as supporting sentences.

The authors have followed the reviewer's instructions.

  1. The reference needs to be enriched from the literature published five years back. MDPI reference is strongly recommended.

The references have been enriched with more literature published in the last five years. Publications with the MDPI reference have also been added.

  1. Please reduce the literature used as a reference that is authored by the present author in order to reduce the number of self-citation.

The bibliography belonging to the author of this manuscript has been reduced, as indicated by the reviewer.

  1. The authors need to proofread the manuscript due to grammatical errors and language style.

The manuscript has been revised avoiding grammatical errors and paying attention to the style of the language.

  1. After peer review, it is encouraged that a graphical abstract be included in the submission.

Figure 6 is the same one that we want to use as a graph for the abstract.

Round 3

Reviewer 3 Report

It is improved, well done.